# Identification and Functional Analysis of Individual-Specific Subpathways in Lung Adenocarcinoma

**DOI:** 10.3390/genes13071122

**Published:** 2022-06-23

**Authors:** Jingya Fang, Zutan Li, Mingmin Xu, Jinwen Ji, Yanru Li, Liangyun Zhang, Yuanyuan Chen

**Affiliations:** 1College of Agriculture, Nanjing Agricultural University, Nanjing 210095, China; 2018201002@njau.edu.cn (J.F.); 2019201003@njau.edu.cn (Z.L.); 2016201007@njau.edu.cn (M.X.); 2020201011@stu.njau.edu.cn (J.J.); 2021201004@stu.njau.edu.cn (Y.L.); 2Department of Mathematics, College of Science, Nanjing Agricultural University, Nanjing 210095, China

**Keywords:** lung cancer, individual-specific subpathways, immunity, biomarkers

## Abstract

Small molecular networks within complex pathways are defined as subpathways. The identification of patient-specific subpathways can reveal the etiology of cancer and guide the development of personalized therapeutic strategies. The dysfunction of subpathways has been associated with the occurrence and development of cancer. Here, we propose a strategy to identify aberrant subpathways at the individual level by calculating the edge score and using the Gene Set Enrichment Analysis (GSEA) method. This provides a novel approach to subpathway analysis. We applied this method to the expression data of a lung adenocarcinoma (LUAD) dataset from The Cancer Genome Atlas (TCGA) database. We validated the effectiveness of this method in identifying LUAD-relevant subpathways and demonstrated its reliability using an independent Gene Expression Omnibus dataset (GEO). Additionally, survival analysis was applied to illustrate the clinical application value of the genes and edges in subpathways that were associated with the prognosis of patients and cancer immunity, which could be potential biomarkers. With these analyses, we show that our method could help uncover subpathways underlying lung adenocarcinoma.

## 1. Introduction

Lung cancer is one of the most malignant tumors. In both male and female patients, lung cancer is the leading cause of cancer-related deaths worldwide [1]. The incidence and mortality of lung cancer have been increasing year by year, with a significant burden on patients and society [2]. In lung cancer, non-small cell lung cancer (NSCLC) accounts for more than 80% of cases, of which lung adenocarcinoma (LUAD) accounts for 45–55% of NSCLC. The 5-year overall survival rate of LUAD is less than 15% [3]. Due to the extensive heterogeneity of LUAD [4], it is crucial to develop successful personalized treatment, which could help in the implementation of precision therapy and improve patient survival rates.

The interactions of multiple dysfunctional genes and pathways can lead to cancer. Pathway analysis is an effective tool for understanding the perturbation or stimulation of biological systems and clarifying the pathogenesis and progression of cancer [5]. In homeostasis, genes coordinate to achieve specific biological functions in the same pathway, and many pathway analysis methods can effectively reveal complex traits and dysregulated pathways in human diseases. Gene set enrichment analysis (GSEA) approaches, which are based on differentially expressed (DE) genes, have been widely used in pathway analysis [6,7]. Similarly, edge set enrichment analysis can identify dysregulated pathways by capturing changes in pathway-specific biological relationships from gene expression data [6]. The concept of subpathways, which are defined as smaller and local molecular networks, has been proposed. Because an entire pathway is too large to accurately interpret as relevant pathological phenomena, a pivotal subpathway region containing fewer components may be more effective and sensitive for dissecting related phenomena [4]. Some studies have reported that dysregulated subpathways are related to the occurrence and development of cancer and can reflect more specific biological functions. Li C et al. identified a “norepinephrine metabolism” subpathway by using pathway topological structure information [7]. This subpathway belongs to a part of the “Tyrosine metabolism” pathway, which is highly associated with cancer development and progression. Han J et al. developed a method, subpathway-GM, based on integrating gene and metabolite information and their topologies within pathways [8]. They identified some subpathways enriched in differentially expressed genes associated with the progression of lung cancer. psSubpathway [8] was proposed to identify phenotype-specific subpathways in a breast cancer dataset. These subpathways could be used as prognostic biomarkers for patients with breast cancer and may contribute to the characterization of patient states [9]. However, these methods mainly focused on groups of tumor or tumor-adjacent samples, rather than considering the potential of this analysis for personalized medicine.

In this study, we aimed to identify reliable individual subpathways that reflect the specific disease state of a patient. Unlike previous studies, genetic changes in the molecular states of a single sample, rather than a group of samples, were integrated to identify significant subpathways associated with the study condition. Specifically, gene interaction information was obtained from gene expression profiles and pathway graphs, and the edges of each sample were constructed for statistically significant edge scores by Pearson correlation coefficients (PCCs). Next, gene set enrichment analysis (GSEA) was applied to each sample based on the edge score to obtain aberrant personalized subpathways. To confirm that our method can sensitively capture relevant biological and clinical information, we applied it to lung cancer datasets from the TCGA database and verified it on the GEO database. Gene mutations and immune-related gene changes from the identified subpathways were used to construct a Cox survival prediction model for the diagnosis and prognosis of patients. Survival analysis suggested the clinical relevance of these subpathway genes, and our results show that the predictive model has an excellent ability to distinguish samples’ survival outcomes. These subpathways were also significantly associated with previously reported functional pathways in LUAD. Thus, we trust that our method is reliable in lung adenocarcinoma prognostic prediction and especially valuable for precision medicine.

## 2. Materials and Methods

### 2.1. Data Preparation

Corresponding TCGA RNA-seq and somatic mutation data were obtained from the UCSC Xena data portal (https://gdc.xenahubs.net (accessed on 15 May 2022)). In this study, the LUAD dataset was used. The gene expression values were normalized to Fragments per Kilobase per Million (FPKM) and were then log_2_(x + 1)-transformed. In total, LUAD tissues and their tumor-adjacent normal tissues were studied, including 526 LUAD tumor samples and 59 LUAD tumor-adjacent tissue samples. Genes with zero expression values in all samples were excluded from the following analysis. Finally, we obtained 49,340 genes. The MuTect2 Variant Aggregation and Masking version was used as somatic mutation data. Additional clinical data of all samples were also downloaded using the above portal to ensure the consistency of the sample source. We removed patients whose survival time or vital status was unspecified.

GSE68465 gene expression profiles were obtained from the NCBI Gene Expression Omnibus (http://www.ncbi.nlm.nih.gov/geo/ (accessed on 15 May 2022)) database as the validation dataset. The probes were mapped to gene symbols, and those that mapped to the same gene symbol were merged by averaging their expression values. Gene expression values were also log_2_-transformed.

### 2.2. Pathway Information

Biological pathway information was obtained from the widely used Kyoto Encyclopedia of Genes and Genomes (KEGG) database [10], which provides abundant pathway information, including interactions, regulation, modifications, and binding between genes. Based on the structure information, each pathway was converted into an undirected graph. Finally, the pathway information was organized into three columns. The first column was the pathway name, and the other two columns were the coding genes connected in the pathway. In total, 75,578 gene pairs (i.e., edges) were obtained as the edge set of 236 pathways.

### 2.3. Overview of the Subpathway Identification Pipeline

This study identified dysregulated subpathways driven by somatic mutation genes based on the construction method of the sample-specific network [11]. A schematic overview is described in Figure 1. Subpathways for each patient were obtained by the following steps: (1) individual-level edge scores in pathways were computed for each patient, which reflects the interactions among genes at the individual biological level; (2) rank-based *edge-score* lists were generated by ranking the edge score of each sample in descending order; then, GSEA [12] was employed to determine if members of an edge set tend to occur toward the top (or bottom) of the edge list, a condition in which the edge set is correlated with the phenotypic class distinction. This approach was usually used to analyze and interpret coordinate pathway-level changes, but here, the gene expression value was replaced with the edge score, and then GSEA was used on each sample to identify individual-level dysregulated subpathways.

### 2.4. Calculation of Edge Scores

As shown in Figure 1, the gene expression data of a group of normal samples served as a reference. The normal samples’ *PCC*s (Pearson correlation coefficients, *PCC_n_*) for each pair of genes in an edge were computed, and then a single sample was added to obtain a new sample (*PCC_n+_*_1_). For each sample, the differential *PCC* (*PCC_n+_*_1_
*− PCC_n_*) between all normal samples and an additional single sample was used in the following analysis.

Due to the specificity of individual samples, different samples have different differential *PCC*s ($\Delta PCC_{n}=PCC_{n+1}-PCC_{n})$ in the same background network. The $\Delta PCC_{n}$ distribution was similar to the normal distribution, which has been theoretically derived in detail [11]. We calculated the edge score, which has a strong theoretical foundation, to quantify each differential edge in the network for a single sample in terms of statistical significance. In fact, the edge score measures the perturbation of edges in a pathway and is defined as follows, where *n* is the total number of reference samples:(1)$$edge-score=\frac{PCC_{n+1\;}-PCC_{n\;}}{(1-PCC_{n}^{2})/\left(n-1\right)}$$

Finally, an *edge-score* matrix was constructed, which contained 75,578 rows (edge numbers) and 526 columns (tumor samples).

### 2.5. Subpathway Extraction

To evaluate the difference in each sample enrichment analysis based on edges, we extracted individual-level subpathways through the following steps (Figure 1).

First, the *edge-score* matrix was used to rank all of the pathway edges of each sample in descending order, and edge lists were generated. Then, enrichment analysis was performed for each sample by using the GSEA-based edge list. We used the “clusterProfiler” package in R/Bioconductor to implement the GSEA analysis. In contrast to differentially expressed genes, which are used in the traditional GSEA method, here, we used the edge score, which reflects the interaction between genes in pathways and is used to measure the difference between different edges by the Pearson correlation coefficients (*PCCs*) of gene pairs. Meanwhile, the original background gene sets were replaced with edge sets in different pathways. In sum, we extracted subpathways based on the results of the GSEA method. The results of core enrichment were the main enrichment edges in different pathways, which had the greatest contribution to the enrichment score of the edge set. The core enrichment edges with FDR < 0.05 in each sample were selected as the set of edges significantly enriched in the corresponding KEGG pathway in each sample, and then all edges were integrated as a subpathway in the corresponding KEGG pathway of a single sample.

### 2.6. Survival Analysis

To further explore the effect of subpathway edges and genes on patient prognosis, the Cox proportional hazards model was applied to each sample. Based on the edge score, the *risk score* of the subpathway for each sample was calculated by the sum of weighted edge scores, where the weight was the regression coefficient from the univariate Cox regression analysis estimated on the edge score and the overall survival data. Based on the mutated gene expression value, the *risk score* of the subpathway for each sample was calculated by the sum of weighted mutated gene expression values, where the weight was the regression coefficient from the univariate Cox regression analysis estimated on the mutated gene expression value and the overall survival data. The *risk score* was defined as:(2)$$risk-score=\overset{k}{\underset{i=1}{\sum }}\beta_{i}Exp\left(i\right)$$
where $\beta_{i}$ is the Cox regression coefficient of edge/mutated gene *i* in a patient; *Exp*(*i*) is the corresponding value of the edge score/mutated gene *i*; and *k* is the number of edges/mutated genes.

We grouped patients using the median of the *risk score* as the cut-off to classify patients into a high-risk group and a low-risk group. Then, a Kaplan–Meier survival analysis was performed for the two groups of patients, and statistical significance was assessed with a significant log-rank test *p*-value < 0.05. The Kaplan–Meier survival curve was utilized to validate the predictive ability of the risk model.

## 3. Results

### 3.1. Mutated Genes in Subpathways

Somatic mutation genes are mutated after conception and are a type of individual-specific information provided by TCGA for every cancer. Driver mutation genes of cancer can provide a growth advantage to cancer cells [13]. To date, 125 driver mutation genes (Appendix A) have been found for human cancer. We used LUAD somatic mutation genes and all driver mutation genes to validate the personalized features of the subpathway in the same sample. TCGA LUAD gene expression data were used to construct individual-specific subpathways, in which the higher the gene degree in a subpathway, the greater its variation/change from normal to tumor samples.

We conducted the computation for the top 5, 10, 20 and 30 highest-degree genes (the genes that have the most connected genes, indicating the importance of the genes) in LUAD subpathways for each patient, and the mean rate at which genes with a high degree are also somatic mutation genes was calculated. We observed that the rate of the top five highest-degree genes was the highest (Figure 2a). Thus, high-degree genes in the subpathway are more likely to have significant mutations in LUAD.

Next, we calculated the mean percentage of the top 5, 10, 20 and 30 highest-degree genes that are both somatic mutation genes and driver mutation genes in subpathways. As shown in Figure 1b, from the top 30 highest-degree genes, the rate monotonically increases, and the higher the gene degree, the higher the likelihood that this gene is a driver mutation gene. The above results indicate that high-degree genes in a subpathway are strongly related to driver mutations in the same sample, and thus, these genes can be used to predict potential driver genes on an individual basis for each sample. The rate at which a somatic mutation gene with a high degree is also a driver mutation gene increases in each subpathway, and thus, the accuracy of the prediction increases with the degree. These results are consistent with the literature [11]. Hence, high-degree genes in the subpathway are more likely to have a significant influence on cancer.

### 3.2. Potential Disease Genes in Subpathways

The high-degree genes in a subpathway are important features of a cancer sample, which reveals their importance in the dysfunctional subpathway of the single sample. They are also strongly related to driver mutation genes in the individual sample (Figure 2b). Therefore, high-degree genes in subpathways may play an important functional role in the subpathway for cancer occurrence.

We collected somatic mutation genes from the top five highest-degree genes in every LUAD sample (a total of 187 top 5 high-degree genes). Genes that appeared in at least 10 samples were selected as potential disease genes (Appendix A) and were further validated by their mutation ratios. The mutation ratios of the above potential disease genes, as well as the mutation ratios of random mutation genes (the same number as potential disease genes), were computed, and the random process was repeated 1000 times. As expected, there was a significantly higher mutation ratio for potential disease genes than random genes in LUAD, as shown in Figure 3.

Then, we examined whether these high-degree potential disease genes could help stratify patients into distinct clusters that were linked to survival. The top five genes with the highest-degree potential for disease genes (Appendix A) were used for survival analysis. We selected seven genes that were significantly associated with patient survival with a *p*-value < 0.05 by univariate Cox regression analysis (Figure 4a). The Kaplan–Meier curve with a log-rank statistical examination was used to perform the survival analysis. As shown in Figure 4b, patients in the low-risk group had considerably better overall survival than those in the high-risk group. These results imply that the mutation of potential disease genes may be used as potential prognostic biomarkers in survival risk stratification in LUAD.

### 3.3. Immune-Related Genes in Subpathways

Tumor tissues have heterogeneous microenvironments (composed of fibroblasts, blood vessels, immune cells and stromal cells), which can be infiltrated by immune cells and affect tumor development. The tumor immune microenvironment is involved in tumorigenesis, progression and metastasis. The interactions between tumor cells and the surrounding infiltrate, especially two major non-tumor constituents (stromal cells and immune cells), can drive either tumor progression or inhibition. We first extracted immune-related genes according to the list of genes curated by the Immunology Database and Analysis Portal (IMMPORT) website [14] and calculated the percentage of the top 5, 10, 20 and 30 highest-degree genes in subpathways that are also immune-related genes. Our results show that the higher the gene degree, the higher the likelihood that this gene is an immune-related gene (Figure 5a).

Aiming to explore the tumor microenvironment of LUAD, we investigated the differences in cancer immunity between high-risk and low-risk groups classified by 28 top 5 high-degree and immune-related genes (Figure 5b). The Cox proportional hazards model was applied to calculate *risk scores* and then stratify patients into two class groups. The ESTIMATE method [15] was applied to calculate the stromal scores and immune scores in LUAD patients. This method predicts the level of infiltrating stromal and immune cells by performing single-sample GSEA on gene expression data. Patients were classified into high- and low-risk groups based on the median of *risk scores*, where the high-risk group had poor survival outcomes (Figure 5c; log-rank *p*-value < 0.0001 in Kaplan–Meier survival analysis). The results indicate that high-degree immune-related genes may be used as potential prognostic biomarkers. Stromal and immune cell infiltration in the tumor immune microenvironment decreases tumor purity. We found that the tumor stromal and immune scores of the high-risk group were significantly lower than those of the low-risk group. This suggests that the tumor purity was high in the high-risk group, which is consistent with the poor survival results of the high-risk group (Figure 5d, Wilcoxon analysis, *p*-value < 2.2 × 10^−16^). These results indicate that significant genes in subpathways may be associated with the tumor immune microenvironment.

### 3.4. Cancer-Related Subpathway Identification

In Section 3.1, we report that mutation genes play an important role in subpathways. Therefore, we selected high-frequency mutation genes (expressed in more than 5% of the samples) for further analysis. Among 17,105 LUAD somatic mutated genes, 683 high-frequency mutated genes were screened. Based on the GSEA results of LUAD patients, the binomial distribution test found 485 significant edges, which were enriched in more than 50% of the samples (*p*-value < 0.05). There were 45 edges containing high-frequency mutation genes among the 485 edges. We then performed a Cox proportional hazard regression model to choose 12 edges that were significantly associated with patient survival with a *p*-value < 0.05 (detailed in Section 2.6). Samples were divided into a high-risk group and a low-risk group based on the median *risk score*.

The results showed that the *risk score* was a risk factor (hazard ratios (HRs) > 1, *p*-value = 4.3 × 10^−6^), while all single edges were not risk factors (hazard ratios (HRs) < 1) (Figure 6a). The *risk score* also had a good classification effect in patients (Figure 6b). These results indicate that high-frequency mutation genes may also affect neighboring genes in subpathways, and these edges could be prognostic biomarkers providing novel insights for patient stratification.

To illustrate the effectiveness of our method, we focused on entire pathways using the binomial distribution test, which found 18 significant pathways with *p*-values < 0.05 that were enriched in more than 50% of the samples (Table 1), all closely related to tumor development and progression.

We focused on a subpathway that belonged to the “cell cycle” (Figure 7a). The key subpathway region was at the bottom of the pathway and centered on *CDK1*. *CDK1* has been reported as a poor prognostic marker of LUAD that is highly correlated with the risk of cancer recurrence and poor overall survival in LUAD patients. Furthermore, *CDK1* has been hypothesized as a potential target for the treatment of LUAD [16]. When comparing *CDK1* expression values between normal and tumor samples in LUAD, the expression value was significantly increased in cancer samples (*p*-value < 2.2 × 10^−16^) (Figure 7b). Interestingly, *CDK1* was also the highest-degree gene in the cell cycle subpathway. In Figure 7c, we show common subnetworks of *CDK1* from some LUAD samples. In addition, the minichromosome maintenance (*MCM*) gene family plays a crucial role in DNA replication and cell cycle progression [17,18]. Several other key nodes, such as *MCM2* dysregulation, are associated with cell proliferation, cell cycle progression and migration [18,19]. Similarly, *MCM4* overexpression is an oncogenic event in LUAD [19]. Kikuchi et al. showed that high expression of *MCM4* was associated with worse overall survival and progression-free survival [20]. Finally, *MCM6* overexpression reduced immune infiltration and response to immunotherapy in LUAD patients [21,22]. Taken together, these results suggest that mutation genes affect the perturbation of edges and lead to the perturbation of subpathways, and that our method can capture important functional components from subpathways that influence the development of cancer.

Another identified functional subpathway belonged to the “Hippo signaling pathway” (Figure 8), which has been verified as an important regulatory mechanism in LUAD. A key subpathway identified was linked to the pathway region of the Wnt signaling pathway, YAP/TAZ, one of the key nodes in this subpathway. Amplification of the *YAP1* gene drives lung cancer brain metastasis [23]. *WWTR1* can promote the progression of lung cancer [24], and *WWTR1* overexpression is closely associated with poor differentiation, poor prognosis and metastasis of non-small cell lung cancer [25]. In addition, the knockout of *WWTR1* in mice can reduce lung cancer metastasis [26]. In summary, the key genes of the identified subpathways have been previously associated with LUAD.

### 3.5. Validation on GEO Dataset

The predictive power of our method was further validated with the independent GSE68465 dataset from GEO (details in Section 2.1). We used seven genes from Section 3.2 and twenty-eight genes from Section 3.3 to respectively calculate *risk scores* and stratify patients into the high-risk group and the low-risk group. As shown in Figure 9a, seven of the top five highest-degree and potential disease genes could classify high- and low-risk groups with significant differences in survival outcomes (*p*-value = 0.0056). Additionally, the high-risk group stratified by 28 top 5 high-degree and immune-related genes showed poor survival outcomes compared to the low-risk group (*p*-value = 0.0081) (Figure 9b). These results verify the effectiveness of these genes in predicting patient survival.

Subpathways were extracted by the same method, and again, a subpathway belonging to the “cell cycle” pathway was selected (Figure 10). The key subpathway region was at the bottom of the pathway and linked to the MAPK signaling pathway. The MAPK pathway has also been reported to play a key role in LUAD [27]. These results are consistent with the results from the TCGA dataset, supporting the effectiveness of our method.

## 4. Discussion

Gene mutation plays an important role in cancer development. In the past decade, large-scale genomic studies have revealed driver genes of LUAD [28]. Several oncogenic drivers have guided novel targeted therapies and immunotherapies against immune checkpoints [29,30,31]. Biological pathway dysfunction plays an important role in tumor occurrence, development and prognosis, which could reflect key cellular mechanisms [10]. Pathway-centered approaches rely on complex molecular interactions and networks and facilitate the identification of robust prognostic features [32]. Subpathways are defined as gene subregions in biological pathways, which contain fewer components than an entire pathway but reflect more specific biological functions. Thus, determining mutation-mediated dysregulated subpathways is important for exploring the pathogenesis of cancer.

In our study, we did not use the predefined gene sets of pathways in GSEA but instead used gene pairs in the pathway. Here, we described the edge score, a computational score that measures the degree of edge interaction instead of individual gene expression values, and identified subpathways by using GSEA for each sample. This method provides a novel approach to subpathway analysis by using edge interactions, which may uncover new insights into the biological system. Based on our results, we concluded that our method can measure the perturbation of a patient’s pathways and be useful for identifying different functional subpathways in individual samples. Furthermore, we demonstrate that the mutation genes in these subpathways play a key role, with a higher degree in the subpathways and biological functions. Finally, we show that these mutation genes can also be used as potential prognostic biomarkers in survival risk stratification of LUAD patients and were related to the patient-specific immune microenvironment, which is conducive to personalized therapy. However, there are a few limitations to our findings: our method only focuses on LUAD and disregards all other cancer types; it is based on a single pathway database (KEGG); and the predictive prognostic gene/edge signatures identified here need to be verified by molecular biological experiments on clinical samples in future studies. Nonetheless, our study may provide a novel method for determining disease-specific functional subpathways and prognostic biomarkers in LUAD.

## 5. Conclusions

In summary, our study could identify and dissect subpathways for individual samples, which provides useful resources for promoting precision cancer therapy and molecular mechanism studies. Our findings unveil a new potential future research direction in personalized immunotherapy.

## Figures and Tables

**Figure 1 genes-13-01122-f001:**
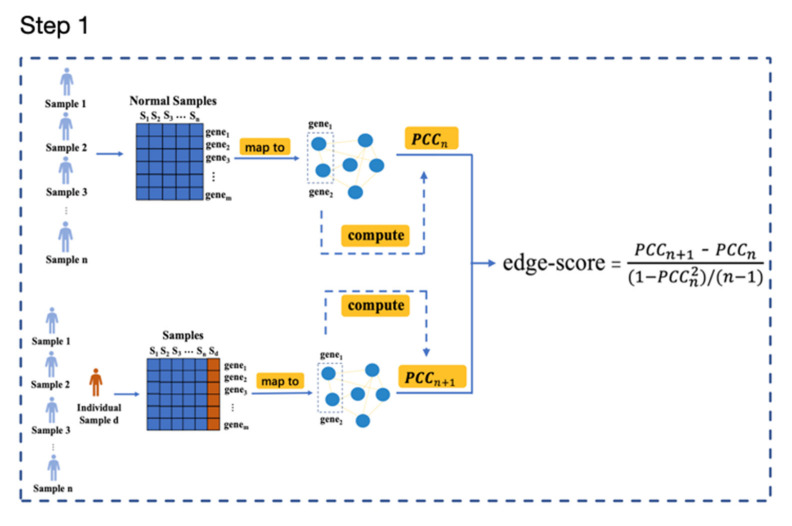
Schematic overview of subpathway extraction.

**Figure 2 genes-13-01122-f002:**
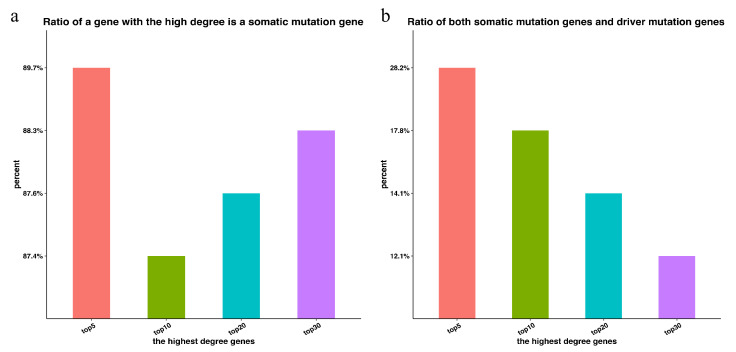
(**a**) The proportion of genes that are somatic mutation genes in the top 5, 10, 20 and 30 highest-degree genes in LUAD subpathways; (**b**) the proportion of genes that are both somatic mutation genes and driver mutation genes in the top 5, 10, 20 and 30 highest-degree genes in LUAD subpathways.

**Figure 3 genes-13-01122-f003:**
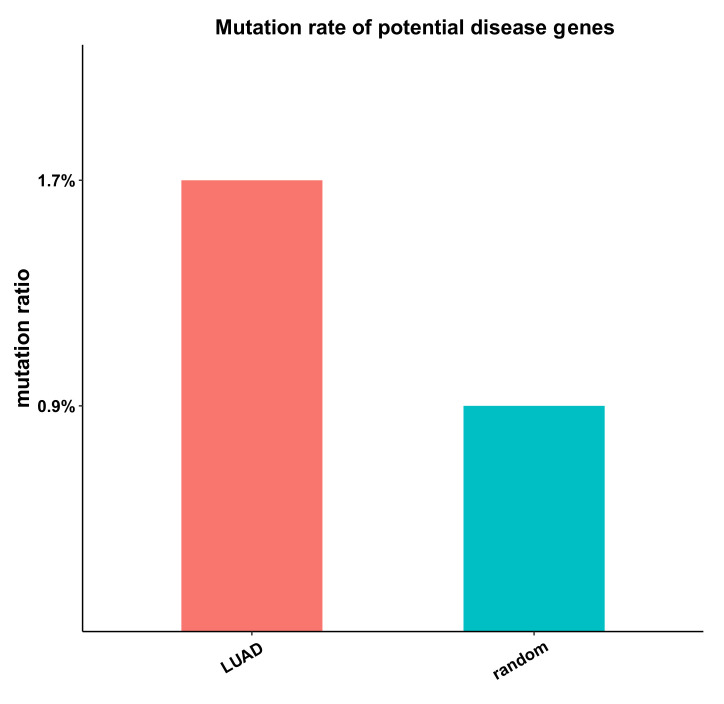
The mutation ratio of potential disease genes (red) and the mutation ratio of random genes (green).

**Figure 4 genes-13-01122-f004:**
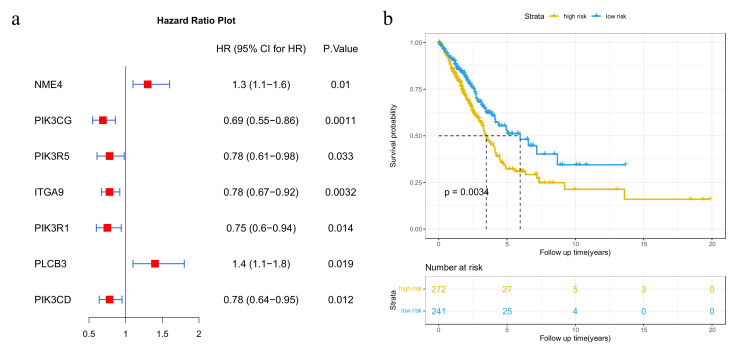
Results of survival analysis of the top 5 genes with the highest-degree potential disease genes. (**a**) Forest plot of 7 genes selected by the univariate Cox proportional hazards regression model; (**b**) Kaplan–Meier survival plots of patients grouped by the median *risk score*.

**Figure 5 genes-13-01122-f005:**
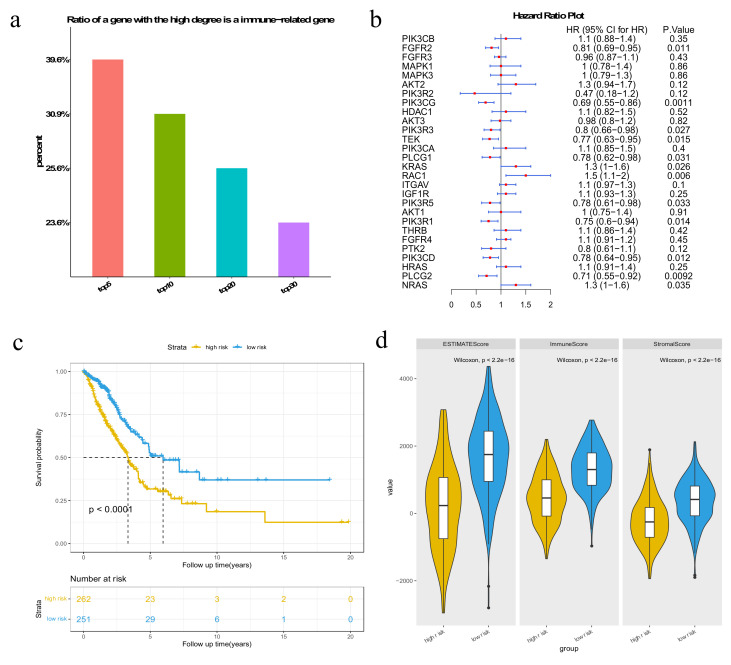
Results of survival analysis of both the top 5 highest-degree genes and immune-related genes. (**a**) The proportion of genes that are immune-related genes in the top 5, 10, 20 and 30 highest-degree genes in subpathways. (**b**) Forest plot of 28 genes selected by the univariate Cox proportional hazards regression model; (**c**) Kaplan–Meier survival curves of patients classified into high-risk and low-risk groups using 28 genes that belong to both the top 5 high-degree and immune-related genes in the subpathway; (**d**) box plot to show the estimate scores, immune scores and stromal scores for the patients in the above two groups.

**Figure 6 genes-13-01122-f006:**
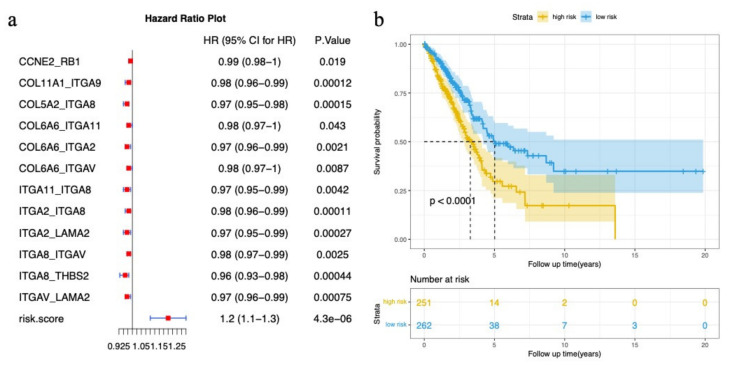
Results of survival analysis of edges. (**a**) Forest plot of 12 edges in LUAD selected by the univariate Cox proportional hazards regression model; (**b**) Kaplan–Meier survival curves of two patient groups.

**Figure 7 genes-13-01122-f007:**
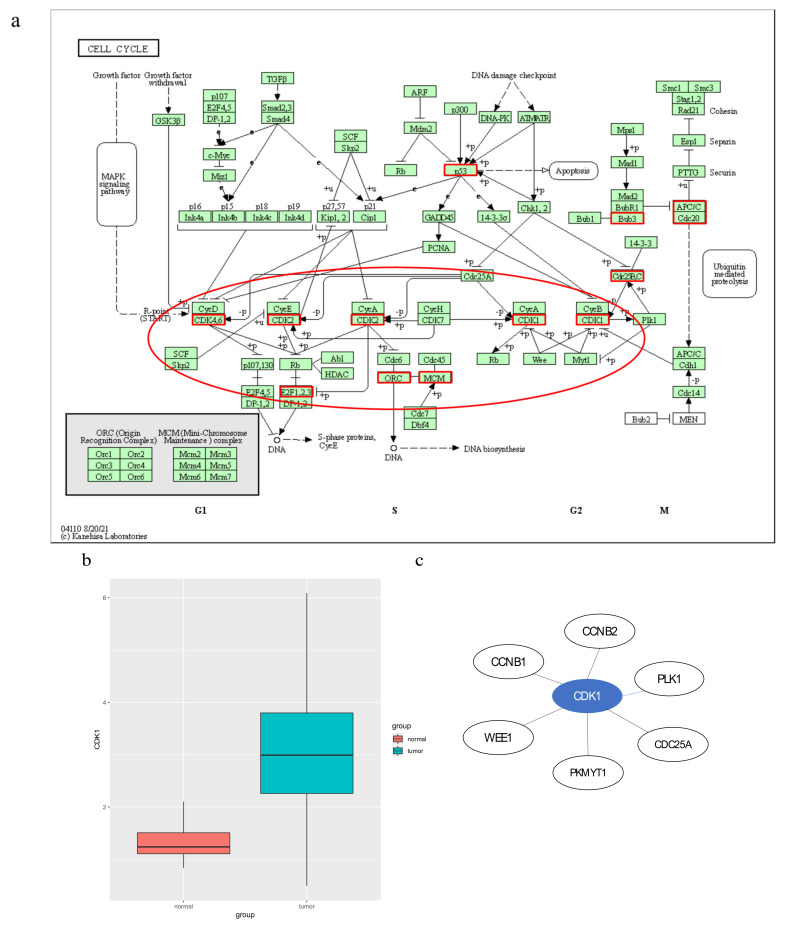
(**a**) The cell cycle pathway in KEGG. Red borders represent genes mapped to the subpathway. Key subpathway region is shown in red ellipse; (**b**) the *CDK1* expression value between normal and tumor samples in LUAD. (**c**) Common subnetworks of *CDK1* from some samples for LUAD.

**Figure 8 genes-13-01122-f008:**
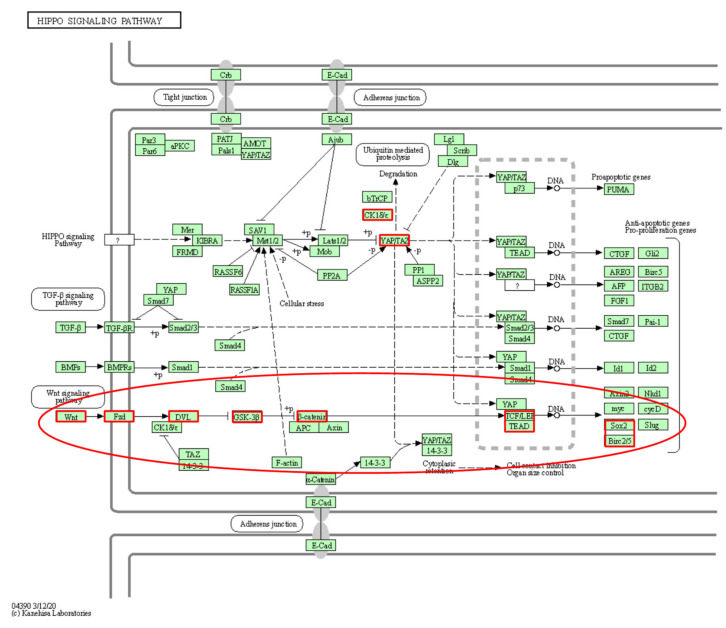
The Hippo signaling pathway in KEGG. Red borders represent genes mapped to the subpathway. Key subpathway region is shown in red ellipse.

**Figure 9 genes-13-01122-f009:**
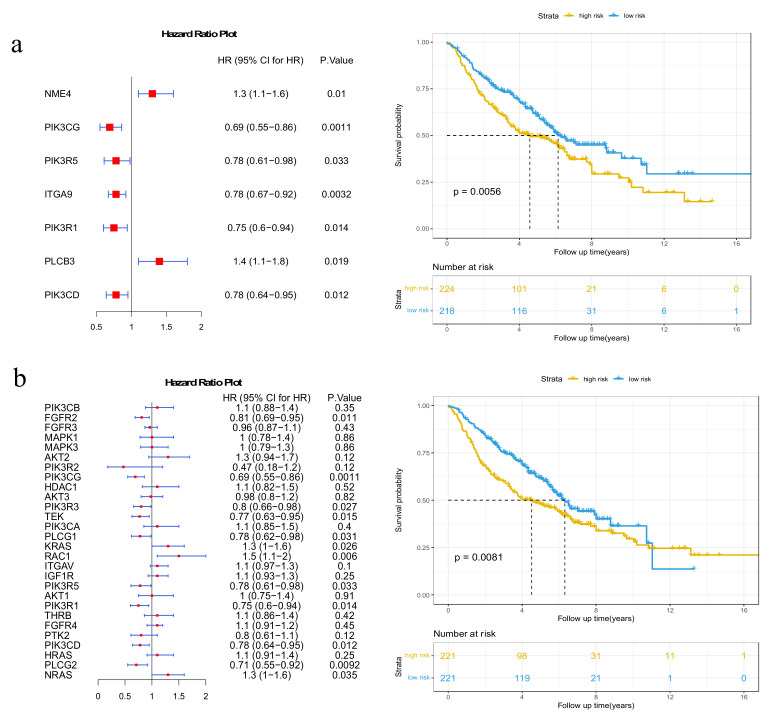
Results of survival analysis in GSE68465. (**a**) Kaplan–Meier survival curves of patients classified into high-risk and low-risk groups using 7 genes that belong to the top 5 genes with the highest-degree potential disease genes; (**b**) Kaplan–Meier survival curves of patients classified into high-risk and low-risk groups using genes that belong to both the 28 top 5 high-degree and immune-related genes.

**Figure 10 genes-13-01122-f010:**
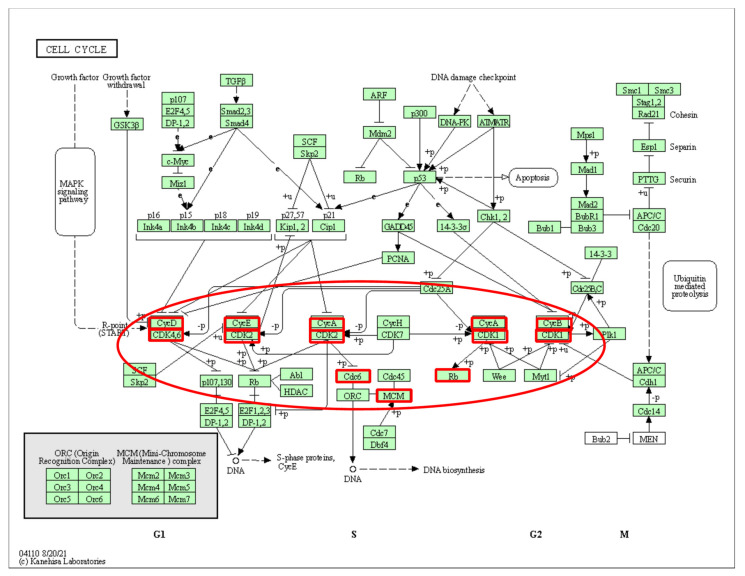
The cell cycle pathway in KEGG. Red borders represent genes mapped to the subpathway from GSE68465. Key subpathway region is shown in red ellipse.

**Table 1 genes-13-01122-t001:** Enriched pathways that were enriched in more than 50% of the samples.

Enriched Pathways	The Number of Enriched Samples
Pyrimidine metabolism	439
Alcoholism	394
Cell cycle	386
Thyroid hormone signaling pathway	386
ECM–receptor interaction	380
Oxidative phosphorylation	363
Hippo signaling pathway	360
Inositol phosphate metabolism	358
Apoptosis	356
Phosphatidylinositol signaling system	355
Small cell lung cancer	353
Wnt signaling pathway	344
Pathways in cancer	322
Basal cell carcinoma	309
Lysine degradation	301
Focal adhesion	300
Axon guidance	286
PI3K-Akt signaling pathway	284

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
