# Peer review of "Identification and Functional Analysis of Individual-Specific Subpathways in Lung Adenocarcinoma"

_genes, 2022, doi:10.3390/genes13071122_

Round 1

Reviewer 1 Report

The findings presented in this manuscript are potentially important and may be of interest to a wide range of biologists as the tool described may find application in many areas.  However, in many places the language used obscures the meaning. While technical terms need to be used to precisely describe for experts the methods used, for the non-specialist (non-bioinformaticians/statisticians) it would be helpful if a number of the terms were explained e.g. 'edge score' and 'highest degree genes' which are used frequently i.e. rather than just explaining how they are derived, it would help if a brief sentence, saying what they are in real terms, were added.  This would give the reader more confidence that the conclusions reached are valid.

The discussion is better from this point of view but the clearer explanations there come too late in the manuscript.

Author Response

Thanks for your insightful comment. In graph theory, the connection between gene and gene as edge (gene pair) and ‘edge score’ in this article is a score to measure the difference between different edges by the Pearson correlation coefficients (PCCs) of gene pairs. To clarify this conception, we added a sentence ‘is a score to measure the difference between different edges by the Pearson correlation coefficients (PCCs) of gene pairs.’ in Line 155-156. ‘Degree’ is also a conception in graph theory, which represents the number of edges directly connected to a node (gene) in the network. Thus, the ‘highest degree genes’ in this article represents the gene that has most connected genes, indicating the importance of the gene. To clarify this conception, we added a sentence ‘(the gene that has most connected genes, indicating the importance of the gene)’ in Line 205-206. We hope our explanation will be make you satisfaction. Thanks for your comment again.

Reviewer 2 Report

Peer review of the manuscript id 1750932 entitled "Identification and functional analysis of individual-specific 2 subpathways in lung adenocarcinoma" received from Genes/MDPI. The authors propose to use a perturbation analysis of gene-gene correlations to identify sub-networks involved in cancer disease. Authors analyze mutation-subnetworks, then immune-related sub-networks, another lung dataset. The analysis could become interesting if many methodological and narrative issues are clarified. The details are shown below.

Major concerns

  • Authors must clarify what is the dimension of the edge-score matrix
  • Edge-Scores will always be obtained for any gene and sample. Nevertheless, the original publication [11] is able to estimate non-random scores. Authors must demonstrate that the gene-scores of the significant GSEA pathways are the result of not random scores (that is, that gene-scores were not low).
  • Authors must demonstrate that the results from Gene Scores are not similar to classical (single sample GSEA) or Gene Expression Z-Scores
  • A negative-control / random experiment is needed. That is an experiment where the same methodology is applied but is expected to provide negative results. For this, a cross-validation-like procedure seems appropriate. Using N-1 normal samples and then the Nth sample to estimate the perturbation. Then all samples, then GSEA, then observe what calls are observed.
  • The subpathway extraction procedure is not clear. That is, it is not clear how a subpathway is "explored" (given all possible combinations of subpathways within a pathway) nor how a subpathway is selected as best/optimal. These concepts or clearer explanation is needed.
  • Is not clear the survival association process followed for the 3 analyses. For example, from the 485 edges, how many genes were screened for survival? The criteria to select 12 were the p-value? Is the p-value corrected for FDR ?
  • The analysis of the GSE68465 dataset is confusing since it is not clear why 7 or 28 genes were "used" since these number does not match with 3.2 and 3.3 sections.

Minor Concerns

  • "path-ways"
  • Reference to the proposal of "subpathways" is missing (introduction)
  • "con-firm"
  • "Ex-pres"
  • Line 110 refers to "Figure10 " which seems to be wrong.
  • Line 123 refers to "Figure11 " which does not exist.
  • Line 138, Eq 1, states "????????? "
  • Is not clear what "o(1/n2)" or "O(1/n^3/2)" are meant to calculate.
  • Some Kaplan-Meier curves show Conf Intervals while others do not. Please show consistent figures to avoid confusion.

Reviewer 3 Report

I recommend minor English revision.

Author Response

Thanks for your comment. We had handed over to professional companies to check this article language before submission. We also made modifications according to your suggestions.